# Evaluation of a Novel Fixative Solution for Liquid-Based Cytology in Diagnostic Cytopathology

**DOI:** 10.3390/diagnostics13243601

**Published:** 2023-12-05

**Authors:** Nadia Casatta, Alessia Poli, Sara Bassani, Gianna Veronesi, Giulio Rossi, Clarissa Ferrari, Carmelo Lupo

**Affiliations:** 1Innovation Department, Diapath S.p.A., Via Savoldini n.71, 24057 Martinengo, Italy; 2Pathology Unit, Fondazione Poliambulanza, Via Bissolati n.57, 25124 Brescia, Italy; 3Research and Clinical Trials Office, Fondazione Poliambulanza Istituto Ospedaliero, Via Bissolati n.57, 25124 Brescia, Italy

**Keywords:** cytopathology, cell fixative, pre-analytical procedures, morphology, gynecological sample, non-gynecological sample, diagnosis

## Abstract

Since its introduction in the early 2000s, liquid-based cytology (LBC) has been increasingly used for gynecologic and non-gynecologic cytology, and its multiple advantages have been widely recognized. The aim of this study was to investigate the use of a new fixative and pre-analytical method for morphological diagnosis in cytological samples. In particular, we evaluated the effect of a novel preservative solution on the preparation of diagnostic slides by comparing it with the standard reference used globally in cytology laboratories. This study included both gynecological (*n* = 139) and non-gynecological (*n* = 183) samples. Several morphologic variables were then identified and evaluated. Using this approach, we were then able to demonstrate the suitability of the new system, with improved safety, to be integrated within current pathology clinical practice. Overall, using a safer preservative solution, the study shows no statistical difference (and then non-inferiority) in the new fixation protocol compared with the standard reference used in routine practice in terms of diagnostic adequacy, evaluated both in clinically relevant gyn and non-gyn datasets.

## 1. Introduction

The proper definition of a diagnosis from different human samples involves a series of steps from correct acceptance to the optimal pre-analytical phase, up to adequate sample preparation for direct examination using a conventional light microscope and/or digitalized images in order to allow the pathologist to obtain the correct visualization of any cell and/or architectural alteration within samples, on which further investigations can possibly be carried out where necessary. Liquid-based cytology (LBC) is increasingly being used both for gynecologic (gyn) and non-gynecologic (non-gyn) cytology. Differently from conventional preparations in which cells are directly smeared on the glass, in LBC cells are rinsed into a liquid preservative collection medium and processed on manual or automated devices. Since its introduction in the early 2000s, LBC has become a useful cytopathological method for diagnosis in various organs [1,2,3,4]. Several advantages were proven, such as the possibility of preparing uniform specimens, avoidance of drying, minor sample loss and a lower percentage of overall unsatisfactory specimens. Thus, the LBC method began to be used widely, especially for gynecological malignancies [5,6,7,8].

The pre-analytical phase, which essentially includes warm ischemia, fixation and tissue processing, is crucial in such processes because it determines the optimal conservation of the sample before subsequent analyses. Numerous studies have in fact demonstrated that sample handling and processing affect the quality of sample morphology, proteins and nucleic acids [9,10,11]. The role of fixative solutions has historically been identified based on the need to maintain tissue or cell samples in conditions as similar as possible to those of the patient at the time of collection. Ideally, a fixative should be non-toxic and economically affordable, and allow good yield in morphological analyses, histochemical and immunohistochemical stains and optimal preservation of all macromolecules to allow subsequent analyses. It is indeed recognized that liquid fixation and the automated filtering procedure may change the original structure of the cells depending on the sampled site and cell type; generally, effusions and washings give fewer interpretative problems than aspiration biopsies since the relationship between the cells and the background is less critical. Effusions in serosal cavities indeed represent an excellent application of the liquid-based technique applied to cytology [12]. Despite this ideal goal, historically each type of fixative has shown the production of a variable number of artifacts, which, over time, pathologists get used to when reading samples for the generation of a diagnosis. For example, the alterations caused on fabrics by formaldehyde-based fixatives are well known [13,14,15].

In cytology, the commonly used preservative solutions are instead alcohol-based, mainly ethanol- or methanol-based [16,17]. Due to its dehydrating properties, methanol denatures proteins, modulating hydrophobic and hydrogen bonds [18]. Ethanol is a historically used fixative both in cytology and in histopathology, known for its reduced effects on nucleic acids and the ability not to influence the conformation of a significant number of antigens [19,20,21].

Another important aspect in the evaluation of different fixative solutions is their toxicity; in addition to the patient themself at the time of sampling, technicians and pathologists who work constantly in contact with fixative solutions can often be highly exposed to unsafe chemicals. The best-known case is certainly formalin, labelled as a type 1B carcinogen [22], whose exposure risks should not be underestimated [23,24,25,26]. In cytology, methanol is also often used, as stated above, although its toxicity on humans was also widely proven [18,27].

The aim of this study was to investigate the use of a new fixative and pre-analytical method for morphological diagnosis in cytological samples. In particular, we evaluated the effect of a novel preservative solution on the preparation of diagnostic slides by comparing it with the standard reference used globally in cytology laboratories. Both gynecological and non-gynecological samples were included in the study. Several morphologic variables were then identified and evaluated to permit a statistical comparison between the gold-standard fixative and the novel fixative solution. By this approach, we were then able to demonstrate the suitability of the new system, with improved safety, to be integrated within current pathology clinical practice.

## 2. Materials and Methods

### 2.1. Ethics Statement and Patients

This study was conducted according to the guidelines of the Declaration of Helsinki, and approved by the Ethics Committee of Brescia, Italy (protocol code NP 5579-STUDIO CYTOPATH^®^, 26 October 2022). Written informed consent was obtained from all patients included in the study.

### 2.2. Specimens

The study comprised a total of 139 gynecological specimens and 183 non-gynecological samples from patients referring to the Fondazione Poliambulanza Hospital (Brescia, Italy), collected from January to May 2023. The flowchart of the method was shown in Figure 1. All gynecological cytology specimens were collected two times and rinsed in the two cell preservation solutions separately. In order to avoid artifacts derived from the experimental procedure (the second sampling may normally be less representative due to a second passage into the uterine cervix, likely leading to the presence of bloody residues), samples were randomly collected by fixing the first sampling material either in PreservCyt^®^ (Hologic Inc., Marlborough, MA, USA) or in Cytopath^®^ solution (Diapath S.p.A., Martinengo, Italy), hereafter referred to as standard and new fixatives, respectively. Non-gynecological samples were derived from different body districts: fluid specimens consisted of unused portions of urine, pleural fluids, FNA, effusions or peritoneal washing. Each specimen was centrifuged for 10 min at 2000 rpm and cell pellets were equally divided into both preserving solutions.

### 2.3. Preparation and Staining

The vials were kept for at least 30 min at RT for proper cell fixation and then processed for cytological assessment. Samples in the PreservCyt^®^ fixative were processed using the TP5000 automated slide preparation system (Hologic Inc., Marlborough, MA, USA), hereafter named Std, standard fixative solution; samples in the Cytopath^®^ fixative were instead processed with Cytopath^®^ processor (Diapath S.p.A., Martinengo, Italy), hereafter named New, new fixative solution. In both cases, the vials were inserted into the automated slide processor, which prepared the cell thin layer. One slide was prepared for each case. The slides were stained with Papanicolaou or Hematoxylin and Eosin (H&E) stain according to standard routine procedure employed in the Fondazione Laboratory of Pathology.

### 2.4. Scoring System

On stained slides, a comparison was made based on the evaluation of 10 different variables to describe the overall quality of the samples, the features more related to the specific method and the effects of the fixative solution on cell preservation. Namely, diagnostic adequacy, overall quality, cellularity, background, cell clusters, cell distribution on spot, cell morphology, cell fixation, staining quality and nuclear details were analyzed. The above characteristics were analyzed by a semi-quantitative scoring system (Table 1). Figure 2 and Figure 3 show representative samples for the endpoints evaluated in this study. Three experienced pathologists blindly read the cases and provided indications of the quality of every slide.

### 2.5. Statistical Analysis

The descriptive analysis of socio-demographic and procedural data was performed with standard statistics: mean, median, standard deviation and interquartile range for continuous quantitative variables; frequency and percentage for categorical variables. For the comparison of the two methods, paired *t*-tests (or corresponding non-parametric Wilcoxon signed-rank tests when appropriate) were applied for continuous variables and paired chi-squared test (McNemar test) for categorical variables.

Any factors that may have an effect on the features of cell preparation were evaluated by generalized linear models (univariate and multiple logistic models for binary data).

The analysis of the features was performed considering the categorical nature of the score evaluation: 0: low/insufficient score; 1: good; 2: excellent score (Table 1). In particular, the scores were dichotomized into two setting: 0 vs. [1 + 2] (i.e., poor vs. adequate-optimal) and 2 vs. [0 + 1] (i.e., optimal vs. non-optimal).

The analyses were carried out separately for gynecological and non-gynecological samples and performed by R statistical software (https://www.r-project.org/ (accessed on 21 September 2023)) [28]. Significance level was set at *p* = 0.05.

## 3. Results

### 3.1. Samples Description

For gynecological samples, *n* = 139 patients were included in this study. Each patient provided two consecutive samples. In order to avoid bias due to the sampling order of the sampling method, part of the first samples were fixed in the new fixative solution and part in the standard reference fixative solution. A substantial difference was indeed observed in the slides obtained with the first and second sampling points, where a higher presence of hematic background was often visible and less cells were spotted on the slides. From a diagnostic viewpoint, the consecutive 139 gynecological samples included 13 positive cases. Namely, 10 cases of atypical squamous cells of undetermined significance (ASC-US, 7.2%) and 3 low-grade squamous intraepithelial lesions (LSIL, 2.15%) were present, while the remaining 91% samples were negative. Both methods allowed the same diagnosis in all the above-mentioned samples. Non-gynecological samples were prepared from the non-diagnostic fluids of patients. Among the 183 samples collected for this study, 3 reported suspected tumor cells, equally distinguishable in samples produced by both methods.

Overall, the differences in morphological features between the cytological samples obtained with the two methods do not appear to be significant (Figure 4), as shown in detail in the paragraphs below.

### 3.2. Gynecological Samples

The following results refer to the gyn dataset. The results reported in Table 2 compare the features between the two approaches proposed by Diapath and Hologic (New and Std, respectively).

Both on the gyn and non-gyn samples, no differences were observed between the preservative solutions in all the features related to the effects of the fixative solution on morphological variables, namely cell morphology, fixation quality, staining and the visibility of nuclear details (Table 2). For this reason, we focused on the features related to the overall process of sample preparation, for which differences between the two methods were observed. Specifically, we compared the two methods on the features assessed on dichotomized score: poor (score = 0) vs. adequate–optimal (score = 1, 2).

In general, no statistical difference was found between the two methods considering the classification of samples as adequate ([1 + 2] categories) vs. non-adequate (category 0); although the background and overall quality of the slide appeared to have higher scores (the best performance) in the new method, the reference one had the best performance in absence of cellular overlapping. As an additional analysis, we performed a comparison considering the classification of samples as excellent (category 2) vs. not-excellent (categories [0 + 1]); the new method had the best performance in spot, adequacy and quality of the cytologic slide (Appendix A). In Table 3, the reported *p*-values are the unadjusted one. However, method comparisons adjusted for diagnosis and sampling sequence were performed and no effects of these two variables were found.

### 3.3. Non-Gynecological Samples

The table below reports the same framework of the above evaluation on the extra-vaginal samples of the non-gyn dataset. The descriptive statistics and comparison considering the endpoint as continuous variables are reported (Table 2). Analyzing the endpoint as a continuous-score scale, the new method performs better than the standard one in spot, diagnostic adequacy, quality of the cytologic slide, cellularity and total score. The standard reference method performed better according to the absence of cell clusters feature.

Using the above features as categories, the two methods were compared (Table 4). Considering the dichotomization of inadequate (0) versus adequate [1 + 2] categories, no statistical differences were observed except for in cellularity, where the standard fixative performed better than the new fixative. Considering the dichotomy of the “excellent” and “not excellent” categories (2 vs. [0 + 1]), no statistical differences were observed for all features except the quality of the cytology slides, where the performance of the new method was better than the standard method (*p* = 0.001; Appendix A).

## 4. Discussion

The two more widespread LBC methods currently in use include filtration-based and centrifugation-based preparations [29]. Both methods were FDA-approved for cervicovaginal cytology in the late 1990s, and since then have also been used for non-gynecological cytology [7,8,30,31]. In parallel, a number of non-FDA-approved methods have been developed for use. LBC has been successfully applied for specimens such as body cavity fluids, urine, cerebrospinal fluid, brushing specimens and FNA of various lesions.

Generally, in the LBC context, specimen materials are collected in specific preservation media and processed normally with automatic systems that provide a slide with a monolayer of cells disposed on the slide. Samples stored in preservation solutions are also routinely used for cell block preparation, which can provide additional diagnostic information, including a more detailed architectural pattern of cell arrangements, and which allow the chance to perform immunocytochemical and/or molecular analysis, as well as special stains. Of relevance, the choice of different fixatives may influence the subsequent analysis: for example, Sato and coworkers found that the suitable fixative type and storage temperature differ depending on antigen location [32].

It is now widely recognized that LBC is superior to conventional preparations for many reasons, including a clearer background, reduced artifacts and confusing extracellular elements, monolayer cell preparation, and the cells being limited to smaller areas with excellent cellular preservation, even if artifacts may be present resulting from the chemical influences of the components of the fixation medium and the physical alterations derived from the processing methods [31,33]. However, as previously mentioned, the effect of every single component in preservative solutions on cellular elements underlines the need of standardizations of the pre-analytical phases in cytological samples. Indeed, for example, the literature shows that alcohol-fixed samples have higher immunoreactivity than those fixed with solutions containing formaldehyde [5,32,34,35]. Similar to other studies [36], this evidence underlines the need for increased standardization and awareness in all of the pre-analytical/analytical steps such as proper collection and preparation of laboratory specimens or adequate fixative/reagent concentrations and technical equipment for every sample treated. So far, very few papers have been entirely focused on the evaluation of the pre-analytical and analytical steps. As reported in [36], the majority of papers do not even mention the pre-analytical and analytical steps or even the controls used.

It is thus essential that those aspects are taken into consideration for the introduction of a new preservative solution and processing method on the market. The present study reports the morphological analysis that was performed on samples treated with a new formulation for the preservation of both gyn and non-gyn samples compared with conventionally used methods. Both methods adequately preserve cellular contents and major morphological aspects in gyn and non-gyn cytology and are thus both adequate for diagnostic purposes. The endpoints chosen in this study are in line with the analyses reported in the literature, where the main aspects for a correct morphological determination of the preservation of a cytological sample are described [7,8]. Only minor differences were observed when treating samples with the new or standard solution. The sampling population was wide enough to be representative of a routine clinical setting with 139 patients recruited for cervical samples and 183 patients for non-gyn samples. Positive cases were also analyzed and no differences in the diagnostic evaluation were referred.

Particular mention should be given to the evaluation of the presence of hematic contaminants and cell distribution on the spot of non-gyn samples: as reported in Figure 5 and Figure 6, a simpler diagnosis can be achieved with the new alternative method, where cells are more homogenously dispersed and blood interference is not present.

An optimal fixative should be nontoxic and cost-effective, and enable a detailed morphological analysis with high-quality histochemical and immunohistochemical staining with preservation of DNA and RNA. Since a fixative with such features is very difficult to obtain, it is essential to explore the most important endpoints in order to ensure the best diagnostic conditions for pathologists. In histology, the use of formalin is linked to the validation and approval of FFPE-dependent tests for the attribution of the best therapy for patients. In the cytology field, the use of formaldehyde is not recommended, with reported advantages of non-cross-linking alcohol-based fixatives, including elimination of carcinogenic vapors and better preservation of macromolecules, thus including both advantages for the sample preservation and for the user’s safety [14,37]. On the basis of the premises given by the results of this study, a parallel pilot investigation is being conducted on the effects of the new fixative method on immunohistochemical and molecular analysis. Preliminary results were presented at the annual congress of the Italian Society of Anatomic Pathology and Cytology [38].

The new fixative evaluated here does not contain either formaldehyde or methanol, providing a safer alternative to the conventionally used method but guaranteeing the possibility to provide the best diagnosis for patients. Overall, with the use of a safer preservative solution, the study demonstrated no statistical difference (and then the non-inferiority) of the new fixation protocol when compared with the standard reference used in routine practice, in terms of diagnostic adequacy evaluated in both wide clinically relevant gyn and non-gyn datasets.

## Figures and Tables

**Figure 1 diagnostics-13-03601-f001:**
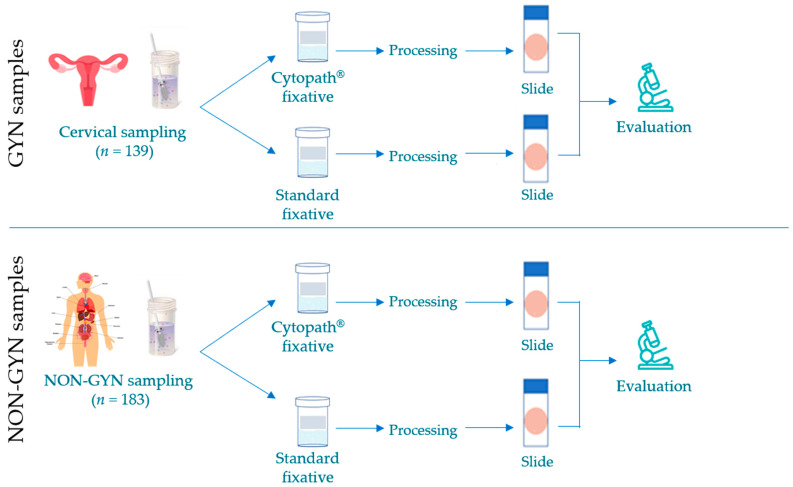
Flowchart showing the methodology followed in this study.

**Figure 2 diagnostics-13-03601-f002:**
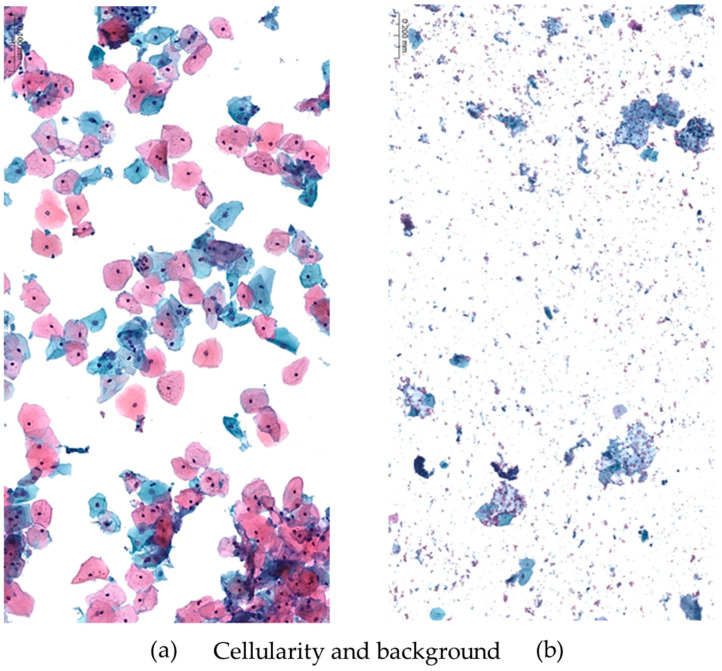
The overall cellularity and the presence of non-cellular debris (necrosis, hemorrhage, amorphous or mucoid material into the background) of the samples were selected among parameters of quality adequacy to compare standard reference and new fixative. On the left (**a**), the cellularity is well dispersed and cell details are easily recognizable. The background is very clean and free from disturbing debris. On the right (**b**), a non-acceptable sample shows a granular background of non-cellular material consisting of necrotic and hemorrhagic debris precluding an optimal visualization of cell details and possibly preventing the correct distribution of the cells along the entire slide. Moreover, a limited number of cells are present that are too isolated in tiny cohesive nests and difficult to examine. (Papanicolaou staining, magnification ×200; gynecologic cytology).

**Figure 3 diagnostics-13-03601-f003:**
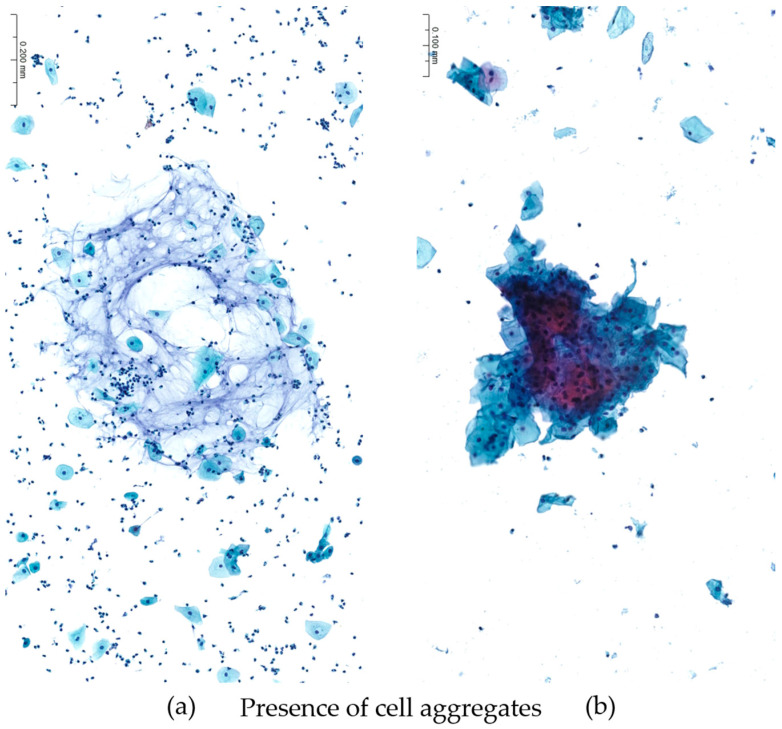
The presence of cellular aggregates in the samples was selected among parameters of quality adequacy to compare standard reference and new fixative. On the left (**a**), despite some focal mucus in the background, non-cellular aggregates precluding the diagnosis are noticed, while on the right (**b**) the sample shows scattered, dispersed cellular aggregates precluding an optimal visualization of cell details. (Papanicolaou staining, magnification ×200; gynecologic cytology.)

**Figure 4 diagnostics-13-03601-f004:**
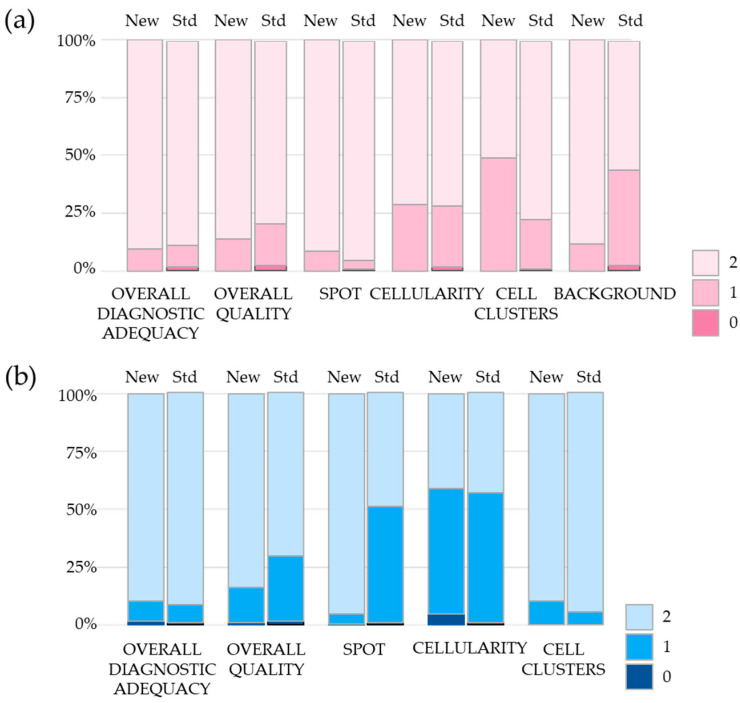
Evaluation of the endpoints on gynecological (**a**) and non-gynecological (**b**) samples. Results are shown as percentage of samples evaluated with score 0, 1 or 2 in gyn and non-gyn samples fixed with the new (New columns) and with standard (Std columns) preservative solutions. Gyn samples: *n* = 139; non-gyn samples: *n* = 183.

**Figure 5 diagnostics-13-03601-f005:**
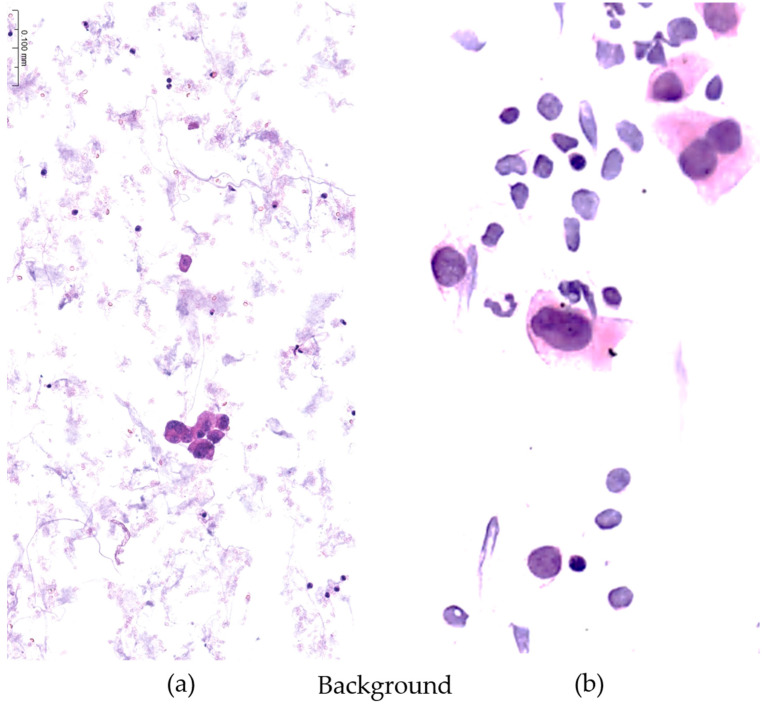
The presence of non-cellular debris (necrosis, hemorrhage, amorphous or mucoid material) ino the background of the sample was also selected among parameters of quality adequacy to compare standard and new fixative). On the left ((**a**) standard reference fixative), a granular background of non-cellular material is shown, consisting of necrotic and hemorrhagic debris precluding an optimal visualization of cell details and possibly hurdling the correct distribution of the cells along the entire slide. On the right (**b**), the sample has a very clean background free from disturbing debris, permitting the correct distribution of the cells along the slide coupled to a better microscopic examination of cell details. (H&E staining, magnification ×200; non-gynecologic cytology from urine.)

**Figure 6 diagnostics-13-03601-f006:**
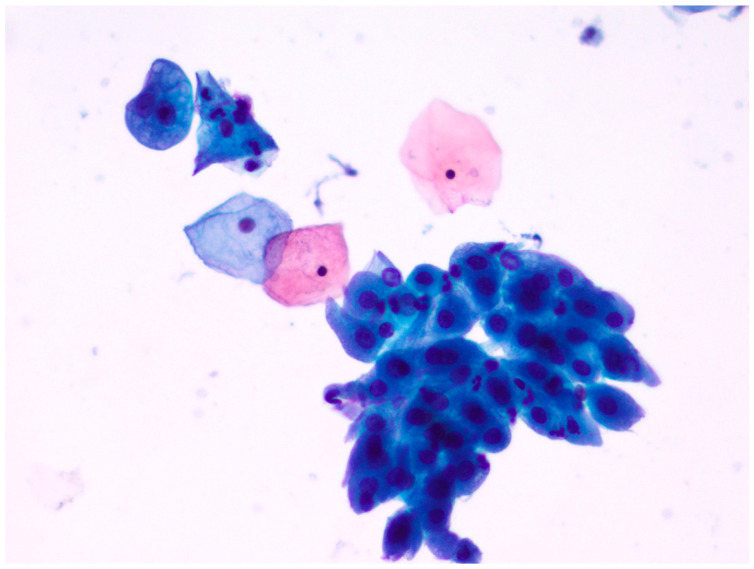
An example of well-dispersed cellularity with some aggregates characterized by clean easily appreciable cellular details (Papanicolaou staining, new fixative, magnification ×200; gynecologic cytology).

**Table 1 diagnostics-13-03601-t001:** Scoring system: criteria used for the analysis of results between the two fixative solutions.

Features		Score		
		0	1	2
General	Diagnostic adequacy	poor	adequate	optimal
	Overall quality	poor	adequate	optimal
Method	Cellularity	insufficient	low	high
	Background (debris)	many confusing elements	partially clean	clean
	Cell clusters	too overlapped	some overlapping	monolayer
	Cell distribution on spot	uneven	partially homogeneous	homogeneous
Fixative solution	Cell morphology	reduced	generally preserved	well preserved
	Cell fixation	poor	adequate	optimal
	Staining quality	not specific	acceptable	specific
	Nuclear details	not visible	good	enhanced

**Table 2 diagnostics-13-03601-t002:** Descriptive statistics of the features on gynecological and non-gynecological samples (as continuous variables with scores 0, 1, 2). Features are grouped according to the characteristics evaluated in the samples: in particular 3 groups were observed, as general, macroscopic and microscopic features. New: new fixative solution; Std: standard fixative solution.

	Gyn Samples			Non-Gyn Samples		
	New	Std	*p*-Value	New	Std	*p*-Value
	(*N* = 139)	(*N* = 139)		(*N* = 183)	(*N* = 183)	
**Overall Diagnostic Adequacy**						
Mean (SD)	1.91 (0.292)	1.88 (0.370)	0.36	1.88 (0.373)	1.90 (0.333)	0.317
Median [Min, Max]	2.00 [1.00, 2.00]	2.00 [0, 2.00]		2.00 [0, 2.00]	2.00 [0, 2.00]	
**Overall Quality Cytologic Slide**						
Mean (SD)	1.86 (0.345)	1.78 (0.467)	0.0384	1.83 (0.409)	1.69 (0.499)	<0.001
Median [Min, Max]	2.00 [1.00, 2.00]	2.00 [0, 2.00]		2.00 [0, 2.00]	2.00 [0, 2.00]	
**Spot**						
Mean (SD)	1.91 (0.282)	1.95 (0.250)	0.251	1.95 (0.251)	1.48 (0.522)	<0.001
Median [Min, Max]	2.00 [1.00, 2.00]	2.00 [0, 2.00]		2.00 [0, 2.00]	1.00 [0, 2.00]	
**Cellularity**						
Mean (SD)	1.71 (0.454)	1.71 (0.488)	0.889	1.36 (0.575)	1.42 (0.517)	0.178
Median [Min, Max]	2.00 [1.00, 2.00]	2.00 [0, 2.00]		1.00 [0, 2.00]	1.00 [0, 2.00]	
**Background**						
Mean (SD)	1.88 (0.320)	1.54 (0.542)	<0.001	1.90 (0.306)	1.95 (0.228)	0.0495
Median [Min, Max]	2.00 [1.00, 2.00]	2.00 [0, 2.00]		2.00 [1.00, 2.00]	2.00 [1.00, 2.00]	
**Cell Clusters**						
Mean (SD)	1.51 (0.502)	1.77 (0.439)	<0.001	8.91 (1.20)	8.44 (1.35)	<0.001
Median [Min, Max]	2.00 [1.00, 2.00]	2.00 [0, 2.00]		9.00 [2.00, 10.0]	8.00 [4.00, 10.0]	
**Cellular Morphology**						
Mean (SD)	2.00 (0)	1.99 (0.120)	0.157	2.00 (0)	2.00 (0)	NA
Median [Min, Max]	2.00 [2.00, 2.00]	2.00 [1.00, 2.00]		2.00 [2.00, 2.00]	2.00 [2.00, 2.00]	
**Fixation Quality**						
Mean (SD)	2.00 (0)	1.99 (0.120)	0.157	2.00 (0)	2.00 (0)	NA
Median [Min, Max]	2.00 [2.00, 2.00]	2.00 [1.00, 2.00]		2.00 [2.00, 2.00]	2.00 [2.00, 2.00]	
**Staining Quality**						
Mean (SD)	2.00 (0)	1.99 (0.120)	0.157	2.00 (0)	2.00 (0)	NA
Median [Min, Max]	2.00 [2.00, 2.00]	2.00 [1.00, 2.00]		2.00 [2.00, 2.00]	2.00 [2.00, 2.00]	
**Nuclear Details Quality**						
Mean (SD)	2.00 (0)	1.99 (0.120)	0.157	2.00 (0)	2.00 (0)	NA
Median [Min, Max]	2.00 [2.00, 2.00]	2.00 [1.00, 2.00]		2.00 [2.00, 2.00]	2.00 [2.00, 2.00]	

**Table 3 diagnostics-13-03601-t003:** Assessment of dichotomized endpoints with categories (0 vs. [1 + 2]) on gynecological samples. New: new fixative solution; Std: standard fixative solution; *: the reported *p*-values are the unadjusted ones.

	New	Std	*p*-Value *
	(*N* = 139)	(*N* = 139)	
**Overall Diagnostic Adequacy**			
0	0 (0%)	2 (1.4%)	0.478
1 + 2	139 (100%)	137 (98.6%)	
**Overall Quality Cytologic Slide**			
0	0 (0%)	3 (2.2%)	0.246
1 + 2	139 (100%)	136 (97.8%)	
**Spot**			
0	0 (0%)	1 (0.7%)	1
1 + 2	139 (100%)	138 (99.3%)	
**Cellularity**			
0	0 (0%)	2 (1.4%)	0.478
1 + 2	139 (100%)	137 (98.6%)	
**Background**			
0	0 (0%)	3 (2.2%)	0.246
1 + 2	139 (100%)	136 (97.8%)	
**Cell Clusters**			
0	0 (0%)	1 (0.7%)	1
1 + 2	139 (100%)	138 (99.3%)	
**TOT_score**			
<12	0 (0%)	4 (2.9%)	0.131
12	139 (100%)	135 (97.1%)	

**Table 4 diagnostics-13-03601-t004:** Assessment of dichotomized features with categories inadequate vs. adequate (0 vs. [1 + 2]) on non-gynecological samples. *: the reported *p*-values are the unadjusted ones. However, method comparison adjusted for diagnosis was performed and no effect of this variable was found.

	New	Std	*p*-Value *
(*N* = 183)	(*N* = 183)
Overall Diagnostic Adequacy			
0	3 (1.6%)	2 (1.1%)	0.998
1 + 2	180 (98.4%)	181 (98.9%)	
Overall Quality Cytologic Slide			
0	2 (1.1%)	3 (1.6%)	0.998
1 + 2	181 (98.9%)	180 (98.4%)	
Spot			
0	1 (0.5%)	2 (1.1%)	0.996
1 + 2	182 (99.5%)	181 (98.9%)	
Cellularity			
0	9 (4.9%)	2 (1.1%)	0.023
1 + 2	174 (95.1%)	181 (98.9%)	
Cell Clusters			
0	0	0	
1 + 2	183 (100%)	183 (100%)	1
TOT_score			
<10	10 (5.5%)	6 (3.3%)	0.343
10	173 (94.5%)	177 (96.7%)

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
