# Peer review of "Evaluation of a Novel Fixative Solution for Liquid-Based Cytology in Diagnostic Cytopathology"

_diagnostics, 2023, doi:10.3390/diagnostics13243601_

Round 1

Reviewer 1 Report

Comments and Suggestions for Authors

Evaluation of a novel fixative solution for Liquid-Based Cytology in diagnostic cytopathology

This is a research paper whereby the authors seek to compare cytological/morphological preservation of a “novel” fixative solution which is not formaldehyde- or methanol-based.  This reviewer does not quite understand what is so “novel” about this preservative and can’t help but to wonder if this “novel” preservative is in fact the CytoPath® Line Cytology Solution.  If it is, it is better to spell out the specifics in both the title and text of the paper.  A search in Pubmed did not reveal any publications mentioning CytoPath® solution, CytoPath® Processor, nor diapath.com.  There is also no mention of any other study using this methodology in the references cited by the authors.  Therefore this is the first paper studying this solution and processor.  The authors need to explain what is so “novel” about this fixative solution other than that it is not formaldehyde-based or methanol-based.  It is understandable the need to guard propriety information but without more information, it is really hard to understand the need to use the word “novel” in the title as well as throughout the text.  It is also hard to imagine why anyone would want to switch from using the time-tested PreservCyt®.  It is speculated that the preservative is likely prepared with ethanol or isopropyl alcohol.  It is hard to argue that these solutions are less toxic than methanol.  None of these preservatives are for human consumption under normal circumstance.

Despite the best intent of the authors, the study involves too few cases.  Yes, using various morphological criteria, the authors show that the “new” solution is not inferior to existing known preservatives which were used millions/billions times over the last twenty five years.  But having so few cases, resulting in a mere 10 cases (7.2%) of ASC-US and 3 cases of LSIL (2.15%), it is hard to interpret usefulness of this “novel” preservative.  It is critically important for any replacement of Hologic ThinPrep® or SurePath® to show equal effectiveness in detecting cytologic abnormalities, especially HSIL and/or cervical squamous or glandular malignancies.  Given that ASC-H/HSIL typically consists of 0.5-1.5% of a screening population, this study needs to have several thousand cases to be able to show equivalency.  Likewise, this study identified 3 cases of suspected malignancies with non-gyn specimens.  This reviewer is not sure that the authors could have conducted a much larger study without the support of the manufacturer.  In fact, the first author of this paper is affiliated/employed by diapath.com.  Perhaps there is a part 2 of the study in bringing this solution to the market.  If it is so, this should be mentioned in the paper.

It is commendable that the authors are studying effect of this new preservative on IHC and molecular analysis.  Once again, it is hard to imagine how the authors are going to conduct an adequate study if sample size continues to be limited.

Ultimately it is good that ThinPrep® and SurePath® have competition.  It “might” help drive down cost of liquid-based cervical cytology screening in less well-to-do communities.  It is hard to establish a competition when FDA requires large amount of data before approving a new processing system and solution.  This reviewer is in favor of having this paper published but would insist that the authors revise the paper to be more forthcoming with the “novelty” of the solution so this paper can become the first step of a long journey in validating a new liquid-based cytology testing system.

Reviewer 2 Report

Comments and Suggestions for Authors

Dear Authors, I congratulate you on the comparative study on two fixation mediums. The manuscript is scientifically sound

Author Response

The authors thank Reviewer #2 for his opinion on the approach used and the structuring of this scientific paper.

Reviewer 3 Report

Comments and Suggestions for Authors

In the current study, the authors evaluated the effect of a novel preservative solution on the preparation of diagnostic slides by comparing it with the stand reference. They found that there was no statistical difference of the new fixation protocol compared with the standard reference.

Overall, the study was well designed with adequate data to support the conclusion.  I have some comments and suggestions as below:

T1. The major concern is the clinical significance of the study. Every preservative has its own advantage and disadvantage. While the data did not show statistical difference (and then non-inferiority) of the new fixation protocol compared with the standard preservative, one may wonder what was the superiority of the new preservative. The author need more discussion to address.

22.  There was no HSIL in the gynecological samples which may raise significant concerns from the readers.

Author Response

First of all, the authors would like to thank Reviewer #3 for taking the time to evaluate this work and for considering the results obtained.

Response 1:

Regarding the comparative evaluation of the Cytopath fixative with the reference in the field, it should be noted that the aim of the study was to validate the entire system for diagnostic purposes. In order to demonstrate the quality of the results obtained, it was decided to propose the study as a comparison with the state of the art with which pathologists and clinicians are usually confronted. The primary objective was therefore not to prove its superiority, but its reliability in providing valid support for diagnosis. In addition, it was emphasised in the text that the formulation was designed to contain neither formaldehyde nor methanol, in order to best protect both the quality of the sample itself and the operator’s health.

Response 2:

The study was designed on such a sample size as to achieve robust statistical significance. In addition, the distinction between gynecological and non-gynecological samples and the analysis of an adequate number of samples for both categories makes the assessments robust and reliable. The samples analyzed were not selected in any way, but derived from the host Institute's diagnostic routine. Samples from all patients who provided consent to participate in the study were included. The fact that we do not feature HSIL in the statistics does not worry the authors about the ability of the preservative solution to act adequately even on positive samples. The evaluation of different morphological parameters in the results of sample preparation using the Cytopath system makes it possible to assess all aspects that may prove critical in the recognition of cellular atypia and thus in the formulation of the best dia